# Tight Dimensionality Reduction for Sketching Low Degree Polynomial Kernels

**Michela Meister**[*]
Cornell University
Ithaca, NY 14850
meister.michela@gmail.com

**Tamas Sarlos**
Google Research
Mountain View, CA 94043
stamas@google.com

**David P. Woodruff**[†]
Department of Computer Science
Carnegie Mellon University
Pittsburgh, PA 15213
dwoodruf@cs.cmu.edu

## Abstract

We revisit the classic randomized sketch of a tensor product of $q$ vectors $x_i \in \mathbb{R}^n$. The $i$-th coordinate $(Sx)_i$ of the sketch is equal to $\prod_{j=1}^{q}\langle u^{i,j}, x^j \rangle / \sqrt{m}$, where $u^{i,j}$ are independent random sign vectors. Kar and Karnick (JMLR, 2012) show that if the sketching dimension $m = \Omega(\epsilon^{-2} C_\Omega^2 \log(1/\delta))$, where $C_\Omega$ is a certain property of the point set $\Omega$ one wants to sketch, then with probability $1 - \delta$, $\|Sx\|_2 = (1 \pm \epsilon)\|x\|_2$ for all $x \in \Omega$. However, in their analysis $C_\Omega^2$ can be as large as $\Theta(n^{2q})$, even for a set $\Omega$ of $O(1)$ vectors $x$.

We give a new analysis of this sketch, providing nearly optimal bounds. Namely, we show an upper bound of $m = \Theta\left(\epsilon^{-2} \log(n/\delta) + \epsilon^{-1} \log^q(n/\delta)\right)$, which by composing with CountSketch, can be improved to $\Theta(\epsilon^{-2} \log(1/(\delta\epsilon)) + \epsilon^{-1} \log^q(1/(\delta\epsilon)))$. For the important case of $q = 2$ and $\delta = 1/\text{poly}(n)$, this shows that $m = \Theta(\epsilon^{-2} \log(n) + \epsilon^{-1} \log^2(n))$, demonstrating that the $\epsilon^{-2}$ and $\log^2(n)$ terms do not multiply each other. We also show a nearly matching lower bound of $m = \Omega(\varepsilon^{-2} \log(1/(\delta)) + \varepsilon^{-1} \log^q(1/(\delta)))$. In a number of applications, one has $|\Omega| = \text{poly}(n)$ and in this case our bounds are optimal up to a constant factor. This is the first high probability sketch for tensor products that has optimal sketch size and can be implemented in $m \cdot \sum_{i=1}^{q} \text{nnz}(x_i)$ time, where $\text{nnz}(x_i)$ is the number of non-zero entries of $x_i$.

Lastly, we empirically compare our sketch to other sketches for tensor products, and give a novel application to compressing neural networks.

## 1 Introduction

Dimensionality reduction, or *sketching*, is a way of embedding high-dimensional data into a low-dimensional space, while approximately preserving distances between data points. The embedded data is often easier to store and manipulate, and typically results in much faster algorithms. Therefore, it is often beneficial to sketch a dataset first and then run machine learning algorithms on the sketched data. This technique has been applied to numerical linear algebra problems [37], classification [9, 10],

---

[*]Work done at Google Research.
[†]Work done at Google Research, and while visiting the Simons Institute for the Theory of Computing.

data stream algorithms [33], nearest neighbor search [22], sparse recovery [12, 20], and numerous other problems.

While effective, in many modern machine learning problems the points one would like to embed are often only specified implicitly. Kernel machines, such as support vector machines, are one example, for which one first non-linearly transforms the input points before running an algorithm. Such machines are much more powerful than their linear counterparts, as they can approximate any function or decision boundary arbitrary well with enough training data. In kernel applications there is a feature map $\phi : \mathbb{R}^n \to \mathbb{R}^{n'}$ which maps inputs in $\mathbb{R}^n$ to a typically much higher $n'$-dimensional space, with the important property that for $x, y \in \mathbb{R}^n$, one can typically quickly compute $\langle \phi(x), \phi(y) \rangle$ given only $\langle x, y \rangle$. As many applications only depend on the geometry of the input points, or equivalently inner product information, this allows one to work in the potentially much higher and richer $n'$-dimensional space while running in time proportional to that of the smaller $n$-dimensional space. Here often one would like to sketch the $n'$-dimensional points $\phi(x)$, without explicitly computing $\phi(x)$ and then applying the sketch, as this would be too slow.

A specific example is the polynomial kernel of degree $q$, for which $n' = n^q$ and $\phi(x)_{i_1, i_2, \ldots, i_q} = x_{i_1} \cdot x_{i_2} \cdots x_{i_q}$. The polynomial kernel is also often used for approximating more general functions via Taylor expansion [17, 30]. Note that the polynomial kernel $\phi(x)$ can be written as a special type of tensor product, $\phi(x) = x \otimes x \otimes \cdots \otimes x$, where $\phi(x)$ is the tensor product of $x$ with itself $q$ times.

In this work we explore the more *general* problem of sketching a tensor product of arbitrary vectors $x^1, \ldots, x^q \in \mathbb{R}^n$ with the goal of embedding polynomial kernels. We will focus on the typical case when $q$ is an absolute constant independent of $n$. In this problem we would like to quickly compute $S \cdot x$, where $x = x^1 \otimes x^2 \otimes \cdots \otimes x^q$, where $S$ is a sketching matrix with a small number $m$ of rows, which corresponds to the embedding dimension.

The most naïve solution would be to explicitly compute $x$ and then apply an off-the-shelf Johnson Lindenstrauss transform $S$ [25, 18, 28, 16], which using the best known bounds gives an embedding dimension of $m = \Theta(\epsilon^{-2} \log(1/\delta))$, which is optimal [24, 27, 31]. However, the running time is prohibitive, since it is at least the number $\text{nnz}(x)$ of non-zeros of $x$, which can be as large as $n^q$. A much more practical alternative is TENSORSKETCH [34, 35] which gives a running time of $\sum_{i=1}^{q} \text{nnz}(x^i)$, which is optimal, but the embedding dimension is a prohibitive $\Theta(\epsilon^{-2}/\delta)$. Note that for high probability applications, where one may want to set $\delta = 1/\text{poly}(n)$, this gives an embedding dimension as large as $\text{poly}(n)$, which since $x$ has length $n^q = \text{poly}(n)$, may defeat the purpose of dimensionality reduction.

Thus, we are at a crossroads; on the one hand we have a sketch with the optimal embedding dimension with a prohibitive running time, and on the other hand we have a sketch with the optimal running time but with a prohibitive embedding dimension. A natural question is if there is another sketch which achieves both a small embedding dimension and enjoys a fast running time.

## 1.1 Our Contributions

### 1.1.1 Near-Optimal Analysis of Tensorized Random Projection Sketch

Our first contribution shows that a previously analyzed sketch by Kar and Karnick for tensor products [30], referred here to as a Tensorized Random Projection, has *exponentially better* embedding dimension than previously known. Given vectors $x^1, \ldots, x^q \in \mathbb{R}^n$ in this sketch one computes the sketch $S \cdot x$ of the tensor product $x = x^1 \otimes x^2 \otimes \cdots \otimes x^q$ where the $i$-th coordinate $(Sx)_i$ of the sketch is equal to $\frac{1}{\sqrt{m}} \cdot \prod_{j=1}^{q} \langle u^{i,j}, x^j \rangle$. Here the $u^{i,j} \in \{-1, 1\}^n$ are independent random sign vectors, and $q$ is typically a constant. The previous analysis of this sketch in [35] describes the sketch as having large variance and requires a sketching dimension that grows as $n^{2q}$, as detailed in the supplementary, in Appendix D.

We give a much improved analysis of this sketch in 2.1, showing that for any $x, y \in \mathbb{R}^{n^q}$ and $\delta < 1/n^q$, there is an $m = \Theta\left(\epsilon^{-2} \log(n/\delta) + \epsilon^{-1} \log^q(n/\delta)\right)$ for which $\Pr[|\langle Sx, Sy \rangle - \langle x, y \rangle| > \epsilon] \leq \delta$. Notably our dimension bound grows as $\log^q(n)$ rather than $n^{2q}$, providing an exponential improvement over previous analyses of this sketch. Another interesting aspect of our bound is that the second term only depends *linearly* on $\epsilon^{-1}$, rather than quadratically. This can represent a substantial savings for small $\epsilon$, e.g., if $\epsilon = .001$. Thus, for example, if $\epsilon \leq 1/\log^{q-1}(n)$, our

sketch size is $\Theta(\epsilon^{-2} \log(n))$ which is optimal for any possibly adaptive and possibly non-linear sketch, in light of lower bounds for arbitrary Johnson-Lindenstrauss transforms [31]. Thus, at least for this natural setting of parameters, this sketch *does not incur very large variance*, contrary to the beliefs stated above. Moreover, $q = 2$ is one of the most common settings for the polynomial kernel in natural language processing [2], since larger degrees tend to overfit. In this case, our bound is $m = \Theta(\epsilon^{-2} \log(n) + \epsilon^{-1} \log^2(n))$, and the separation of the $\epsilon^{-2}$ and $\log^2(n)$ terms in our sketching dimension is especially significant.

We next show in 2.2 that a simple composition of the Tensorized Random Projection with a *CountSketch* [14] slightly improves the embedding dimension to $m = \Theta(\epsilon^{-2} \log(1/(\delta\epsilon)) + \epsilon^{-1} \log^q(1/(\delta\epsilon)))$ and works for all $\delta < 1$. Moreover, we can compute the entire sketch (including the composition with CountSketch) in time $O(\sum_{i=1}^q m \cdot \mathrm{nnz}(x^i))$. This makes our sketch a "best of both worlds" in comparison to the Johnson-Lindenstrauss transform and TensorSketch: Tensorized Random Projection runs much faster than the Johnson-Lindenstrauss transform and it enjoys a smaller embedding dimension than TensorSketch. Additionally, we are able to show a nearly matching $m = \Omega(\varepsilon^{-2} \log(1/\delta) + \varepsilon^{-1} \log^q(1/\delta))$ lower bound for this sketch, by exhibiting an input $x$ for which $\|Sx\|_2 \notin (1 \pm \epsilon)\|x\|_2$ with probability more than $\delta$.

It is also worthwhile to contrast our results with earlier work in the data streaming community [23, 11] that analyzed the variance only for $q = 2$ and general $q$ respectively, and then achieved high probability bounds by taking the median of multiple independent copies of $S$. The non-linear median operation makes the former constructs unsuitable for machine learning applications. In contrast, we show high probability bounds for the *linear* embedding $S$ directly. Recent work [4], which was a merger of [5, 29], provide different sketches with different trade-offs. Their main focus is a sketching dimension with a (polynomial) dependence on $q$, making it more suitable for approximating high-degree polynomial kernels. Our focus is instead on improving the analysis of an existing sketch, which is most useful for small values of $q$.

From a technical standpoint, our work builds off the recent proof of the Johnson-Lindenstrauss transform in [16]. We write the sketch $S$ as $\sigma^T A \sigma$, where in our setting $\sigma$ corresponds to the concatenation of $u^{1,1}, u^{2,1}, \ldots, u^{m,1}$, while $A$ is a random matrix which depends on all of $u^{1,j}, u^{2,j}, \ldots, u^{m,j}$ for $j = 2, 3, \ldots, q$. Following the proof in [16], we then apply the Hanson-Wright inequality to upper bound the $w$-th moment $\mathbf{E}[|\sigma^T A \sigma - \mathbf{E}[\sigma^T A \sigma]|^w]$, for integers $w$, in terms of the Frobenius norm $\|A\|_F$ and operator norm $\|A\|_2$ of the matrix $A$. The main twist here is that in the tensor setting, when we try to apply this inequality, the *matrix $A$ is a random variable itself*. Bounding $\|A\|_2$ can be accomplished by essentially viewing $A$ as a $(q-1)$-th order tensor, flattening it $q - 1$ times, and applying Khintchine's inequality each time. The more complicated part of the argument is in bounding $\|A\|_F$, which again involves an inductive argument to obtain tail bounds on the Frobenius norm of each of the blocks of $A$, which itself is a block-diagonal matrix with $m$ blocks. The tail bounds are not as strong as sub-Gaussian or even sub-exponential random variables, which makes standard analyses based on moment generating functions inapplicable. We instead give a "level-set" argument by giving a novel adaptation of analyses of Tao, originally needed for showing concentration of $p$-norms for $0 < p < 1$, to our tensor setting (see, e.g., Proposition 6 in [36]).

### 1.1.2 Approximating Polynomial Kernels

Replicating experiments from [35], we approximate polynomial kernels using Tensorized Random Projection, TensorSketch, and Random Maclaurin [30] features. In Section 4.1 we demonstrate that TensorSketch always fails for certain sparse inputs, while Tensorized Random Projection succeeds with high probability. We show in 4.2 that Tensorized Random Projection has similar accuracy to TensorSketch, and both vastly outperform Random Maclaurin features.

### 1.1.3 Compressing Neural Networks

We also experiment with using Tensorized Random Projection to compress the layers of a neural network. In [8], Arora et al. propose a method for compressing the layers of a neural network via random projections and prove generalization bounds for such networks. To compress an individual layer, they choose a basis set of random Rademacher matrices and project the layer's weight matrix onto this random basis set. We refer to this method here as *Random Projection*. The simplest, order $q = 2$, Tensorized Random Projection can be viewed as a more efficient, rank-1 version of Random Projection: instead of using a basis set of fully-random Rademacher matrices, the basis set is made

up of random rank-1 Rademacher matrices. We show in 4.3 that Tensorized Random Projection has similar test accuracy as Random Projection when compressing the top layer of a small neural network.

## 1.2 Preliminaries

For a survey of using sketching for algorithms in randomized numerical linear algebra, we refer the reader to [37]. We give a brief background here on several concepts related to our work.

There are many variants of the Johnson-Lindenstrauss Lemma, though for us the most useful is that for an $m \times n$ matrix $S$ of independent entries drawn from $\{-1/\sqrt{m}, 1/\sqrt{m}\}$, if $m = \Omega(\epsilon^{-2} \log(1/\delta))$, then for any fixed vector $x \in \mathbb{R}^n$, we have:

$$\Pr_S[\|Sx\|_2^2 = (1 \pm \epsilon)\|x\|_2^2] \geq 1 - \delta.$$

This lemma is also known to hold for any matrix $S$ with independent sub-Gaussian entries.

The matrix $S$ is dense, and the CountSketch transform is instead much sparser.

**Definition 1.1** (CountSketch). *A CountSketch transform is defined to be $\Pi = \Phi D \in \mathbb{R}^{m \times n}$. Here, $D$ is an $n \times n$ random diagonal matrix with each diagonal entry independently chosen to be $+1$ or $-1$ with equal probability, and $\Phi \in \{0,1\}^{m \times n}$ is an $m \times n$ binary matrix with $\Phi_{h(i),i} = 1$ and all remaining entries 0, where $h : [n] \to [m]$ is a random map such that for each $i \in [n]$, $h(i) = j$ with probability $1/m$ for each $j \in [m]$. For a matrix $A \in \mathbb{R}^{n \times d}$, $\Pi A$ can be computed in $O(nnz(A))$ time, where $nnz(A)$ denotes the number of non-zero entries of $A$.*

We now define a tensor product and various sketches for tensors.

**Definition 1.2** ($\otimes$ product for vectors). *Given $q$ vectors $u_1 \in \mathbb{R}^{n_1}$, $u_2 \in \mathbb{R}^{n_2}$, $\cdots$, $u_q \in \mathbb{R}^{n_q}$, we use $u_1 \otimes u_2 \otimes \cdots \otimes u_q$ to denote an $n_1 \times n_2 \times \cdots \times n_q$ tensor such that, for each $(j_1, j_2, \cdots, j_q) \in [n_1] \times [n_2] \times \cdots \times [n_q]$,*

$$(u_1 \otimes u_2 \otimes \cdots \otimes u_q)_{j_1, j_2, \cdots, j_q} = (u_1)_{j_1}(u_2)_{j_2} \cdots (u_q)_{j_q},$$

*where $(u_i)_{j_i}$ denotes the $j_i$-th entry of vector $u_i$.*

We now formally define TensorSketch:

**Definition 1.3** (TensorSketch [34]). *Given $q$ vectors $v_1, v_2, \cdots, v_q$ where for each $i \in [q]$, $v_i \in \mathbb{R}^{n_i}$, let $m$ be the target dimension. The TensorSketch transform is specified using $q$ 3-wise independent hash functions, $h_1, \cdots, h_q$, where for each $i \in [q]$, $h_i : [n_i] \to [m]$, as well as $q$ 4-wise independent sign functions $s_1, \cdots, s_q$, where for each $i \in [q]$, $s_i : [n_i] \to \{-1, +1\}$.*

*TensorSketch applied to $v_1, \cdots, v_q$ is then CountSketch applied to $\phi(v_1, \cdots, v_q)$ with hash function $H : [\prod_{i=1}^q n_i] \to [m]$ and sign functions $S : [\prod_{i=1}^q n_i] \to \{-1, +1\}$ defined as follows:*

$$H(i_1, \cdots, i_q) = h_1(i_1) + h_2(s_2) + \cdots + h_q(i_q) \pmod{m},$$

*and*

$$S(i_1, \cdots, i_q) = s_1(i_1) \cdot s_2(i_2) \cdot \cdots \cdot s_q(i_q).$$

*Using the Fast Fourier Transform, TensorSketch$(v_1, \cdots, v_q)$ can be computed in $O(\sum_{i=1}^q (nnz(v_i) + m \log m))$ time.*

The main sketch we study is the classic randomized sketch of a tensor product of $q$ vectors $x_i \in \mathbb{R}^n$. The $i$-th coordinate $(Sx)_i$ of the sketch is equal to $\prod_{j=1}^q \langle u^{i,j}, x^j \rangle / \sqrt{m}$, where $u^{i,j}$ are independent random sign vectors. Kar and Karnick show [30] that if the sketching dimension $m = \Omega(\epsilon^{-2} C_\Omega^2 \log(1/\delta))$, where $C_\Omega$ is a certain property of the point set $\Omega$ one wants to sketch, then with probability $1 - \delta$, $\|Sx\|_2 = (1 \pm \epsilon)\|x\|_2$ for all $x \in \Omega$. However, in their analysis $C_\Omega^2$ can be as large as $\Theta(n^{2q})$, even for a set $\Omega$ of $O(1)$ vectors $x$.

## 2 Main Theorem and its Proof

Our main theorem combining sketches $S$ and $T$ described in Sections 2.1 and 2.2 is the following. We provide its proof in Section 2.3.

**Theorem 2.1.** *There is an oblivious sketch $S \cdot T : \mathbb{R}^{n^q} \to \mathbb{R}^m$ for $m = \Theta(\epsilon^{-2}\log(1/(\epsilon\delta)) + \epsilon^{-1}\log^q(1/(\epsilon\delta))$, such that for any fixed vector $x \in \mathbb{R}^{n^q}$ and constant $q$, $\Pr[\|STx\|_2^2 = (1 \pm \epsilon)\|x\|_2^2] \geq 1 - \delta$, where $0 < \epsilon, \delta < 1$. Further, if $x$ has the form $x = x^1 \otimes x^2 \otimes \cdots \otimes x^q$ for vectors $x^i \in \mathbb{R}^n$ for $i = 1, 2, \ldots, q$, then the time to compute $STx$ is $O(\sum_{i=1}^{q} nnz(x^i)m)$.*

## 2.1 Initial Bound on Our Sketch Size

We are ready to present Tensorized Random Projection sketch $S$ and the outermost layer of its analysis. We defer statements and proofs of some key technical lemmas to Appendix A in the supplementary. Note that both the sketching dimension $m$ and the failure probability $\delta$ depend on $n$, which we later eliminate with the help of Section B.

**Theorem 2.2.** *Define oblivious sketch $S : \mathbb{R}^{n^q} \to \mathbb{R}^m$ for $m = \Theta(\epsilon^{-2}\log(n/\delta) + \epsilon^{-1}\log^q(n/\delta))$ as follows. Choose $m \cdot q$ independent uniformly random vectors $u^{i,j} \in \{+1, -1\}^n$, where $i = 1, \ldots, m$ and $j = 1, \ldots, q$. Let the $\ell = 1, \ldots, m$-th row of $S$ be $(1/\sqrt{m})u^{\ell,1} \otimes u^{\ell,2} \otimes \cdots \otimes u^{\ell,q}$, that is, the $(i_1, i_2, \ldots, i_q)$-th entry of the $\ell$-th row of $S$ is $(1/\sqrt{m})\prod_{j=1}^{q} u_{i_j}^{\ell,j}$. Then for any fixed vector $x \in \mathbb{R}^{n^q}$ and failure probability $\delta < 1/n^q$ it holds that $\Pr[\|Sx\|_2^2 = (1 \pm \epsilon)\|x\|_2^2] \geq 1 - \delta$.*

*Proof.* It suffices to show for any unit vector $x \in \mathbb{R}^{n^q}$, that

$$\Pr[|\|Sx\|_2^2 - 1| > \epsilon] \leq \delta. \tag{1}$$

We define $S^i \in \mathbb{R}^{m \times n^{q-1}}$ to have $\ell$-th row equal to $(1/\sqrt{m})u_i^{\ell,1} \cdot v^\ell$, where $v^\ell = u^{\ell,2} \otimes u^{\ell,3} \otimes \cdots \otimes u^{\ell,q}$, and define $x = (x^1, \ldots, x^n)$, with each $x^i \in \mathbb{R}^{n^{q-1}}$, so that $Sx = \sum_{i=1}^{n} S^i x^i$. Then,

$$\|Sx\|_2^2 = \|\sum_{i=1}^{n} S^i x^i\|_2^2 = \sum_{i=1}^{n} \|S^i x^i\|_2^2 + 2\sum_{i \neq i'} \langle S^i x^i, S^{i'} x^{i'}\rangle.$$

Lemma 2.3 below proves that $\sum_{i=1}^{n} \|S^i x^i\|_2^2 = (1 \pm \epsilon/3)\|x\|_2^2$ holds with probability at least $1 - \delta/10$. We prove Lemma 2.3 and in effect Theorem 2.2 by induction on $q$ and applying Theorem 2.2 for $q' = q - 1$. To complete the proof, we need to show that that

$$\sum_{i \neq i'} \langle S^i x^i, S^{i'} x^{i'}\rangle \leq \varepsilon/3 \tag{2}$$

with probability at least $1 - 9\delta/10$. Note that $(S^i x^i)_\ell$, the $\ell$th coordinate of $S^i x^i$, is $(1/\sqrt{m})u_i^{\ell,1}\langle v^\ell, x^i\rangle$. So showing (2) is equivalent to showing $\frac{1}{m}\sum_{i \neq i'}\sum_{\ell=1}^{m} u_i^{\ell,1} u_{i'}^{\ell,1}\langle v^\ell, x^i\rangle\langle v^\ell, x^{i'}\rangle \leq \epsilon/3$. Rearranging the order of summation, we need to upper bound

$$Z := \frac{1}{m}\sum_{\ell=1}^{m}\sum_{i \neq i'} u_i^{\ell,1} u_{i'}^{\ell,1}\langle v^\ell, x^i\rangle\langle v^\ell, x^{i'}\rangle := u^T A u,$$

where $u \in \mathbb{R}^{nm \times 1}$ and $A \in \mathbb{R}^{nm \times nm}$ is a block-diagonal matrix with $m$ blocks, each of size $n \times n$. Let $\mathcal{E}$ be the event that $\sum_{i=1}^{n} \|S^i x^i\|_2^2 = (1 \pm \epsilon/3)$. By Lemma 2.3, we have that $\Pr[\mathcal{E}] \geq 1 - \delta/10$. Furthermore, let $\mathcal{F}$ be the event that $\|A\|_2 = O(\frac{\log^{(q-1)}(qn^q m/\delta)}{m})$ and

$$\|A\|_F = O(1/\sqrt{m} + \log^{1/2}(1/\delta)\log^{(2q-3)/2}(m/\delta)\log\log(m/\delta)/m)$$

bounds hold for the operator and Frobenius norm of $A$. By a union bound over Lemmas A.4 and A.7, we have that $\Pr[\mathcal{F}] \geq 1 - \delta/10$. Lemma A.3 uses the Hanson-Wright Theorem to bound $Z$ in terms of $\|A\|_2$ and $\|A\|_F$ and proves that $\Pr[Z \geq \varepsilon/3 | \mathcal{F}] \leq \delta/2$.

Putting this all together, we achieve our initial bound on $\|Sx\|_2^2$: Taking the probability over all $u^\ell$ and $v^\ell$, we have,

$$
\begin{aligned}
\Pr[|\|Sx\|_2^2 - 1| > \epsilon] &\leq \Pr[\neg\mathcal{E}] + \Pr[|\|Sx\|_2^2 - 1| > \epsilon \mid \mathcal{E}] \\
&\leq \delta/10 + \Pr[Z \geq \epsilon/3 \mid \mathcal{E}] \\
&\leq \delta/10 + \frac{\Pr[Z \geq \epsilon/3]}{\Pr[\mathcal{E}]} \\
&\leq \delta/10 + \frac{\Pr[Z \geq \epsilon/3]}{1 - \delta/10} \\
&\leq \delta/10 + (1 + \delta/5)\Pr[Z \geq \epsilon/3] \\
&\leq \delta/10 + (1 + \delta/5)(\Pr[Z \geq \epsilon/3 \mid \mathcal{F}] + \Pr[\neg\mathcal{F}]) \\
&\leq \delta/10 + (1 + \delta/5)(\delta/2 + \delta/10) = 3\delta^2/25 + 7\delta/10 \\
&\leq 3\delta/25 + 7\delta/10 \leq \delta.
\end{aligned}
$$

From the $\delta \leq 1$ assumption it follows that $\delta^2 \leq \delta$, which implies the second to last inequality and concludes the proof. $\qquad\square$

**Lemma 2.3.** *For all $q \geq 2$, any set of fixed vectors $x^1, \ldots, x^n \in \mathbb{R}^{n^{q-1}}$, sketching dimension $m = \Theta(\epsilon^{-2}\log(n/\delta) + \epsilon^{-1}\log^{q-1}(n/\delta))$, $\delta < 1/n^{q-1}$, and matrices $S^i \in \mathbb{R}^{m \times n^{q-1}}$ defined in the proof of Theorem 2.2, we have that $\Pr[\sum_{i=1}^n \|S^i x^i\|_2^2 = (1 \pm \epsilon/3)\|x\|_2^2] \geq 1 - \delta/10$.*

*Proof.* Define matrix $S_0 \in \mathbb{R}^{m \times n^{q-1}}$ such that its $\ell$-th row is $v^\ell/\sqrt{m}$ from the proof of Theorem 2.2. Additionally define $m \times m$ diagonal matrices $D^i$ such that $D^i_{\ell,\ell} := u_i^{\ell,1}$. Note that $S^i = D^i S_0$ and therefore $\|S^i x^i\|_2 = \|D^i S_0 x^i\|_2 = \|S_0 x^i\|_2$ holds since $D^i$ is $\pm 1$ diagonal matrix. To prove the lemma, it is sufficient to show that

$$\forall i \in [1, n] : \Pr[\|S_0 x^i\|_2^2 = (1 \pm \epsilon/3)\|x^i\|_2^2] \geq 1 - \delta/(10n) \tag{3}$$

holds, since then we have that $\sum_{i=1}^n \|S^i x^i\|_2^2 = \sum_{i=1}^n \|S_0 x^i\|_2^2 = (1 \pm \epsilon/3)\sum_i^n \|x^i\|_2^2 = (1 \pm \epsilon/3)\|x\|_2^2$ with probability at least $1 - \delta$ by a union bound.

We prove inequality (3) by induction on $q$. In the base $q = 2$ case, entries of $v^\ell = u^{\ell,2}$ vectors are i.i.d. $\pm 1$ random variables. Equivalently the entries of $S_0$ are i.i.d. $\pm 1$ random variables. Applying the Johnson-Lindenstrauss lemma [31] to $S_0$ and each $x^i$ with $\delta' = \delta/(10n)$ proves the base case.

Now assume that Theorem 2.2 holds for $q' = q - 1$. Observe that the structure of $S_0$ for $q' = q - 1$ is exactly like that of $S$ for $q$. Setting $\delta' = \delta/(10n)$ in Theorem 2.2 we have that inequality (3) holds for sketching dimension $m' = \Theta\left(\epsilon^{-2}\log(n/\delta') + \epsilon^{-1}\log^{q-1}(n/\delta')\right)$. Since $\log(n/\delta') = \log(n^2/\delta) = \Theta(\log(n/\delta))$ we can simplify $m'$ to $\Theta\left(\epsilon^{-2}\log(n/\delta) + \epsilon^{-1}\log^{q-1}(n/\delta)\right)$ as claimed. $\qquad\square$

## 2.2 Optimizing Our Sketch Size

We define the sketch $T$, which is a tensor product of CountSketch matrices. We compose our sketch $S$ from Section 2.1 with $T$ in order to remove the dependence on $n$. See Section B for the proof.

**Theorem 2.4.** *Let $T$ be a tensor product of $q$ CountSketch matrices $T = T^1 \otimes \cdots \otimes T^q$, where each $T^i$ maps $\mathbb{R}^n \to \mathbb{R}^t$ for $t = \Theta(q^3/(\epsilon^2\delta))$. Then for any unit vector $x \in \mathbb{R}^{n^q}$, we have $\Pr[|\|Tx\|_2^2 - 1| > \epsilon] \leq \delta$. Furthermore, if $x$ is of the form $x^1 \otimes x^2 \otimes \cdots \otimes x^q$, for $x^i \in \mathbb{R}^n$ for $i = 1, 2, \ldots, q$, then $Tx = T^1 x^1 \otimes \cdots \otimes T^q x^q$, where $nnz(T^i x^i) \leq nnz(x^i)$ and where the time to compute $T^i x^i$ is $O(nnz(x^i))$ for $i = 1, 2, \ldots, q$.*

## 2.3 Proof of Theorem 2.1

Finally we prove our main claim by composing sketches $S$ and $T$ from Sections 2.1 and 2.2.

*Proof.* Our overall sketch is $S \cdot T$, where $S$ is the sketching matrix of Section 2.1, with sketching dimension $m = \Theta(\epsilon^{-2}\log(t/\delta) + \epsilon^{-1}\log^q(t/\delta))$, and $T$ is the sketching matrix of Section 2.2, with sketching dimension $t = \Theta(q^3/(\epsilon^2\delta))$. To satisfy the conditions of Theorem

2.2, set $\delta_S = 0.5/t^q$. $S$ is applied with approximation error $\epsilon/2$ and failure probability $\delta_S$ and $T$ is applied with $\epsilon/2$ and $\delta/2$ respectively. Note that $\delta_S \leq \delta/2$ and for $q$ constant, $\log(t/\delta_S) = \Theta(\log(t^{q+1})) = \Theta(\log(t)) = \Theta(\log(1/(\epsilon\delta)))$ holds. Thus, the sketching dimension $m$ of $ST$ is now $\Theta(\epsilon^{-2}\log(1/(\epsilon\delta)) + \epsilon^{-1}\log^q(1/(\epsilon\delta)))$, and has no dependence on $n$. By Theorems 2.2, 2.4, and a union bound, we have that for any unit vector $x \in \mathbb{R}^{n^q}$, $\Pr[|\|S \cdot Tx\|_2^2 - 1| > \epsilon] \leq \delta$.

In Theorem 2.4 above we show that, if $x$ is a vector of the form $x^1 \otimes x^2 \otimes \cdots \otimes x^q$, for $x^i \in \mathbb{R}^n$ for $i = 1, 2, \ldots, q$, then $Tx = T^1x^1 \otimes \cdots \otimes T^qx^q$ where each $T^ix^i$ can be computed in $O(\mathrm{nnz}(x^i))$ time and where $\mathrm{nnz}(T^ix^i) \leq \mathrm{nnz}(x^i)$. Thus, we can apply $S$ to $Tx$ in $O(\sum_{i=1}^q \mathrm{nnz}(x^i)m)$ time.

$\square$

## 3 Lower Bound on Our Sketch Size

We next show that our sketching dimension of $m = \Theta(\epsilon^{-2}\log(1/(\delta\epsilon)) + \epsilon^{-1}\log^q(1/(\delta\epsilon)))$ is nearly tight for our particular sketch $S \cdot T$. We will assume that $q$ is constant. Note that $S \cdot T$ is an oblivious sketch, and consequently by lower bounds for any oblivious sketch [24, 27, 31], one has that $m = \Omega(\epsilon^{-2}\log(1/\delta))$. More interestingly, we show a lower bound of $m = \Omega(\epsilon^{-1}\log^q(1/\delta))$ summarized in the following theorem; see Section C for the proof.

**Theorem 3.1.** *For any constant integer $q$, there is an input $x \in \mathbb{R}^{n^q}$ for which if the number $m$ of rows of $S$ satisfies $m = o(\varepsilon^{-2}\log(1/\delta) + \varepsilon^{-1}\log^q(1/\delta))$, then with probability at least $\delta$, $\|STx\|_2^2 > (1+\epsilon)\|x\|_2^2$.*

Recall that the upper bound on our sketch size, for constant $q$, is $m = O(\varepsilon^{-2}\log(1/(\epsilon\delta)) + \varepsilon^{-1}\log^q(1/(\epsilon\delta)))$, and thus our analysis is nearly tight whenever $\log(1/(\epsilon\delta)) = \Theta(\log(1/\delta))$. This holds, for example, whenever $\delta < \epsilon$, which is a typical setting since $\delta = 1/\mathrm{poly}(n)$ for high probability applications.

## 4 Experiments

We evaluate Tensorized Random Projections in three different applications. In Section 4.1 we show that Tensorized Random Projections always succeed with high probability while TensorSketch always fails on extremely sparse inputs. Then in Section 4.2 we observe that TensorSketch and Tensorized Random Projections approximate non-linear SVMs with polynomial kernels equally well. Finally in Section 4.3 we demonstrate that Random Projections and Tensorized Random Projections are equally effective in reducing the number of parameters in a neural network while Tensorized Random Projections are faster to compute. To the best of our knowledge this comprises the first experimental evaluation of [8]'s compression technique in terms of accuracy. The code for the experiments is available at `https://github.com/google-research/google-research/tree/master/poly_kernel_sketch`.

### 4.1 Success Probability of TensorSketch vs Tensorized Random Projection

In this section we demonstrate that TensorSketch cannot approximate the polynomial kernel $\kappa(x, y) = \langle x, y \rangle^q$ accurately for all pairs $x, y \in V$ simultaneously if the vectors in the set $V$ are not smooth, i.e., if $\|x\|_\infty/\|x\|_2 = \Omega(1)$ holds for all $x$ in $V$. TensorSketch fails even if the sketching dimension $m$ is much larger than $|V|$. On the contrary, Tensorized Random Projection works well.

Let a set $S$ of data points be a standard basis in $d$ dimensions. If $k \geq 2$ coordinates of different vectors collide in the same TensorSketch hash bucket then their common bucket is either zero or non-zero. If it is 0, then $\langle e_i, e_i \rangle^q$ is incorrectly estimated as 0 instead of 1. If the common bucket's value is not 0, then the estimate of $\langle e_i, e_j \rangle^q$ is non-zero, where $i$ and $j$ are any pair of two colliding coordinates. Thus if there is a collision, then TensorSketch cannot estimate all dot products exactly. Moreover the estimate cannot be close to the true kernel value either since if the dot product is incorrect, then it is off by at least 1. Now if $n \geq \sqrt{2m\ln(1/(1-p))}$ then by the Birthday paradox [1] we have at least one collision with probability $p$. If the number of vectors (and dimension) $n$ is greater than the sketching dimension $m$, which is the interesting case for sketching, then there is always a collision by the pigeonhole principle. We remark that [26] provides a more detailed analysis of this sketching

dimension vs input vector smoothness tradeoff for CountSketch, which is a key building block of TensorSketch.

We illustrate the above phenomena in Figure 1(a) as follows. We fix the sketch size $m = 100$ and vary the input dimension (= number of vectors) $n$ along the x-axis. We measure the largest absolute error in approximating $\kappa(e_i, e_j) = \langle x, y \rangle^2 = \delta_{ij}$ among the first $n$ standard basis vectors and repeat the experiment with 100 randomly drawn TensorSketch and Tensorized Random Projection instances. The y-axis shows the average of the maximum error in approximating the true kernel, where error bars correspond to one standard deviation. It is clear that TensorSketch's error quickly becomes the largest possible, 1, as the number $n$ of vectors passes the critical threshold $\sqrt{100}$, while Tensorized Random Projection's max error is much smaller, more concentrated, and grows at a much slower rate in the same setting.

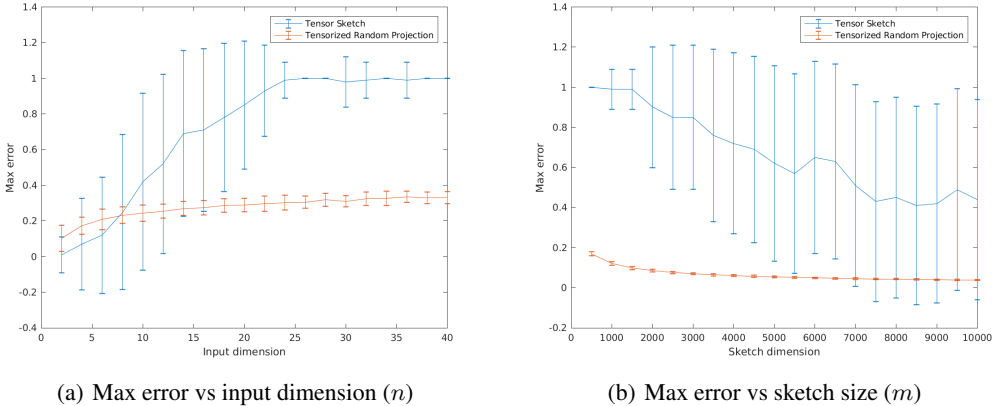

(a) Max error vs input dimension $(n)$        (b) Max error vs sketch size $(m)$

Figure 1: Maximum Error

Next, in Figure 1(b) we fix the input dimension (= number of vectors) to $n = 100$ and vary the sketch size $m$ along the x-axis instead. The y-axis remains unchanged. We again observe that TensorSketch's max error decreases very slowly and it is still about 40% of the largest error possible (1) on average at sketching size $m = n^2 = 10^4 \ll d$. Tensorized Random Projection's max error is almost an order of magnitude smaller at the same sketch size.

## 4.2 Comparison of Sketching Methods for SVMs with Polynomial Kernel

We replicate experiments from [35] to compare Tensorized Random Projections with TensorSketch (TS) and Random Maclaurin (RM) sketch. We approximate the polynomial kernel $\langle x, y \rangle^2$ for the Adult [19] and MNIST [32] datasets, by applying one of the above three sketches to the dataset. We then train a linear SVM on the sketched dataset using LIBLINEAR [21], and report the training accuracy. This accuracy is the median accuracy of 5 trials. Our baseline is the training accuracy of a non-linear SVM trained with the exact kernel by LIBSVM [13]. We experiment with between 100 and 500 random features.

Both Figures 2(a) and 2(b) show that Tensorized Random Projection has similar accuracy to TensorSketch, and both have far better accuracy than Random Maclaurin. Recall that Random Maclaurin approximates the kernel function $\kappa$ with its Maclaurin series. For each sketch coordinate it randomly picks degree $t$ with probability $2^{-t}$ and computes degree-$t$ Tensorized Random Projection. This is rather inefficient for the polynomial kernel, which has exactly one non-zero coefficient in its Maclaurin expansion. Random Maclaurin's generality is not required for the polynomial kernel and we can obtain more accurate results for general kernels by sampling degree $t$ proportional to its Maclaurin coefficient.

## 4.3 Compressing Neural Networks

We begin with a standard 2-layer fully connected neural network trained on MNIST [32] with a baseline test accuracy of around 0.97. The first layer has dimension (784x512) and the top layer has

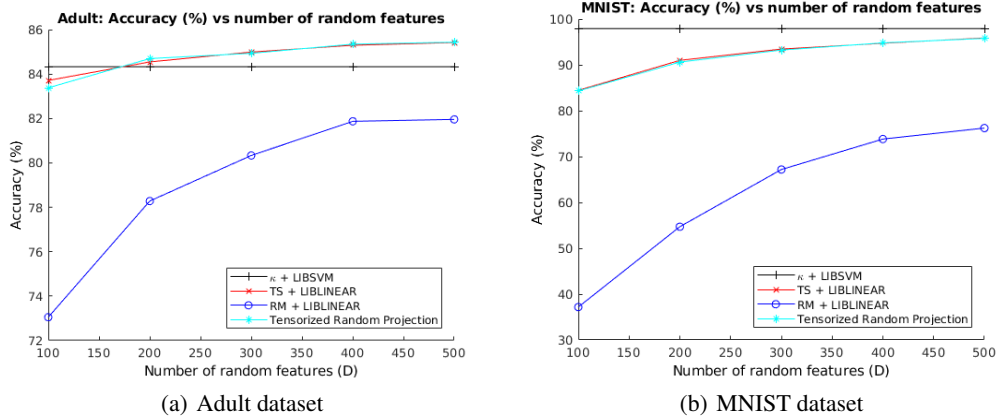

(a) Adult dataset                    (b) MNIST dataset

Figure 2: Accuracy vs Number of Random Features

dimension (512x10). Further specifics of the model can be found in the TensorFlow tutorials [3]. We sketch the weight matrix in the top layer using either Tensorized Random Projection or Random Projection. We then reinsert this sketched matrix into the original model and evaluate its accuracy on the MNIST test set. We compare both the test accuracy and the time needed to compute the sketch for both methods.

In Figure 3(a) we see that both Tensorized Random Projection and Random Projection reach similar test accuracy for the same number of parameters. Figure 3(b) in illustrates that Tensorized Random Projection runs somewhat faster than ordinary Random Projection.

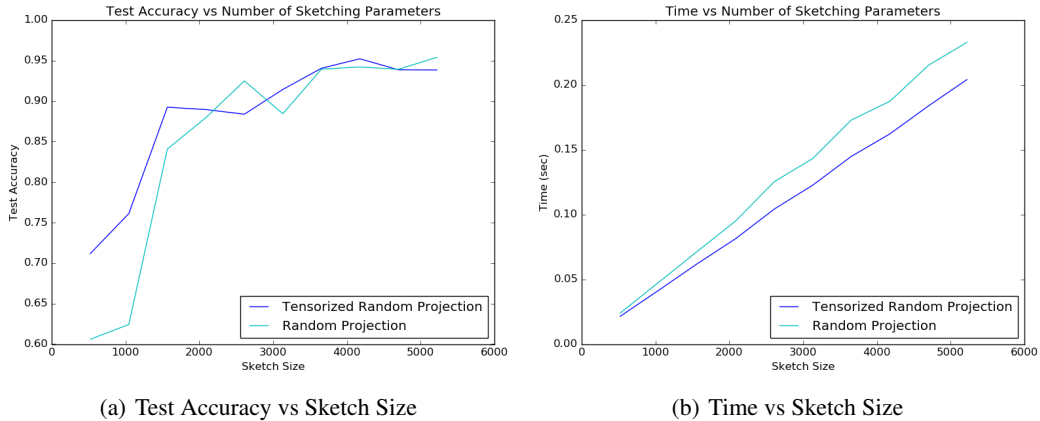

(a) Test Accuracy vs Sketch Size          (b) Time vs Sketch Size

Figure 3: Sketching the Last Layer of MNIST Neural Network

# 5 Conclusion

We presented a new analysis of Tensorized Random Projection, providing nearly optimal bounds and demonstrated its versatility in multiple applications. An interesting question left for future work is whether its $m \cdot \sum_{i=1}^{q} \text{nnz}(x_i)$ running time could be further improved for dense $x$. We conjecture that the iid random $u_i^\ell$ Rademacher vectors might be replaced with fast pseudo-random rotations, perhaps a product of one or more randomized Hadamard matrices similar to ideas in [7], which could possibly lead to an $O(m \log n)$ running time.

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
