[Supplementary Material]

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

# A  Additional Lemmas from Section 2.1

## A.1  Preliminaries

We write $f(x) \lesssim g(x)$ if $f(x) = O(g(x))$. For random variable $X$ and $w \in \mathbb{R}$, $\|X\|_w$ denotes $(\mathbf{E}[|X|^w])^{1/w}$. Minkowski's inequality shows that $\|\cdot\|_w$ is a norm if $w \geq 1$. We use $\|A\|_F$ for the Frobenius norm of a matrix $A$, and $\|A\|$ for its operator norm.

We need the following form of Khintchine's inequality [15].

**Theorem A.1.** *(Khintchine) There is a constant $C > 0$ such that for $\sigma_1, \ldots, \sigma_n$ independent Rademacher (i.e., uniform in $\{1, -1\}$) random variables, and any fixed $x \in \mathbb{R}^n$, $\Pr[(\sum_{i=1}^n \sigma_i x_i)^2 > C\sqrt{\log(1/\delta)}\|x\|_2^2] \leq \delta$.*

We next state the following version of the Hanson-Wright inequality [15] that we need.

**Theorem A.2.** *(Hanson-Wright) For $\sigma_1, \ldots, \sigma_n$ independent Rademachers (i.e., uniform in $\{1, -1\}$) and $A \in \mathbb{R}^{n \times n}$, for all $w \geq 1$,*

$$\|\sigma^T A \sigma - \mathbf{E}[\sigma^T A \sigma]\|_w \leq O(1) \cdot (\sqrt{w}\|A\|_F + w \cdot \|A\|).$$

## A.2  Lemmas

**Lemma A.3.** *Let $Z := u^T A u$, where $A$ and $u$ are as described in Theorem 2.2. Let $\mathcal{F}$ be the event that the bounds on $\|A\|_F$ and $\|A\|_2$ hold as described in Lemmas A.7 and A.4. Then $\Pr[Z \geq \varepsilon/3 | \mathcal{F}] \leq \delta/2$.*

*Proof.* We will set $w = \Theta(\log(1/\delta))$. By Hanson-Wright and the triangle inequality, where the randomness is *taken only with respect to $u^1, \ldots, u^m$ and not with respect to $v^1, \ldots, v^m$*, we have

$$
\begin{aligned}
\|Z\|_w &\leq \|\sqrt{w} \cdot \|A\|_F + w \cdot \|Ax\|\|_w & (4) \\
&\leq \sqrt{w} \cdot \|\|A\|_F\|_w + w \cdot \|\|A\|\|_w. & (5)
\end{aligned}
$$

Note that here we use that for any fixing of $v^1, \ldots, v^m$, and thus $A$, since $i \neq i'$ we have $\mathbf{E}[Z] = 0$, where the expectation is taken with respect to $u^1, \ldots, u^m$. Lemma A.4 proves that, with probability at least $1 - \delta/20$,

$$\|A\|_2 = O\left(\frac{\log^{(q-1)}(qn^q m/\delta)}{m}\right) \tag{6}$$

and Lemma A.7 proves that, with probability at least $1 - \delta/20$,

$$\|A\|_F = O(1/\sqrt{m} + \log^{1/2}(1/\delta)\log^{(2q-3)/2}(m/\delta)\log\log(m/\delta)/m) \tag{7}$$

Define $\mathcal{F}$ to be the event that (6) and (7) both hold. By a union bound, we have that $\Pr[\mathcal{F}] \geq 1 - \delta/10$. We can now bound $\Pr[Z \geq \epsilon/3 \mid \mathcal{F}]$:

$$
\begin{aligned}
\Pr[Z \geq \epsilon/3 \mid \mathcal{F}] &= \Pr[Z^w \geq (\epsilon/3)^w \mid \mathcal{F}] \\
&\leq (\epsilon/3)^{-w}\mathbf{E}[Z^w \mid \mathcal{F}] \\
&\leq (\epsilon/3)^{-w} \cdot 2^w \cdot \mathbf{E}[\max(\sqrt{w}^w \cdot \|A\|_F^w, w^w \cdot \|A\|^w) \mid \mathcal{F}]
\end{aligned}
$$

where we now justify these inequalities. The first inequality is Markov's inequality. The second inequality is the Hanson-Wright inequality, where we have used that $a + b \leq 2\max(a, b)$.

We now set $w = \Theta(\log(1/\delta))$ for a large enough constant in the $\Theta(\cdot)$ notation. Recall that given $\mathcal{F}$, both (7) and (6) hold. We choose $m$ so that it satisfies the following three constraints: We need $m = \Omega(\log(1/\delta)\log^{(2q-3)/2}(m/\delta)\log\log(m/\delta))/\epsilon$ as well as $m = \Omega(\epsilon^{-2}\log(1/\delta))$, so that $\sqrt{w}^w \cdot \|A\|_F^w \le \epsilon^w/C_1^w$. We also need $m = \Omega(\epsilon^{-1}\log^{(q-1)}(nm/\delta)\log(1/\delta))$ so that $w^w \cdot \|A\|^w \le \epsilon^q/C_1^q$.

A sketch size of $m = \Theta\left(\epsilon^{-2}\log(n/\delta) + \epsilon^{-1}\log^q(n/\delta)\right)$ satisfies these constraints, using that $\log(1/\delta)\log^{(2q-3)/2}(m/\delta)\log\log(m/\delta) \le \log^q(n/\delta)$. Note that we can assume $m \le n^q$, otherwise we could instead use the identity matrix as our sketching matrix. With this setting of $m$, we have that

$$\Pr[Z \ge \epsilon/3 \mid \mathcal{F}] \quad \le \quad (\epsilon/3)^{-w} \cdot 2^w \cdot \mathbf{E}[\max((\varepsilon/C_1)^w, (\varepsilon/C_1)^q \mid \mathcal{F}],$$

and by setting $C_1 > \max((2\cdot 6^w/\delta)^{1/w}, (2\cdot 6^w\varepsilon^{q-w}/\delta)^{1/q})$ we have that $\Pr[Z > \varepsilon/3|\mathcal{F}] \le \delta/2$. $\quad\square$

**Lemma A.4.** *With probability at least $1 - \delta/20$, $\|A\|_2 = O(\frac{\log^{(q-1)}(qn^qm/\delta)}{m})$*

*Proof.* Since $A$ is block-diagonal, its operator norm is the largest operator norm of any block. The $\ell$-th block in $A$ can be written as $(1/m)y^\ell(y^\ell)^T$ where $y^\ell \in \mathbb{R}^n$ for each $\ell \in [m]$, and $y_i^\ell = \langle v^\ell, x^i \rangle$. Since $y^\ell(y^\ell)^T$ is a rank-1 matrix, the eigenvalue of the $\ell$-th block is equal to $(1/m)Tr(y^\ell(y^\ell)^T)$, which is at most $(1/m)\|y^\ell\|_2^2$. Thus, $\|A\| \le (1/m)\max_{\ell=1}^m \|y^\ell\|_2^2$ with probability 1, where the probability is taken only over $u^1, \ldots, u^m$.

We next bound $|\langle v^\ell, x^i \rangle|$, where recall $v^\ell = u^{\ell,2} \otimes u^{\ell,3} \otimes \cdots \otimes u^{\ell,q}$. By A.5 we have that $|\langle v^\ell, x^i \rangle| = O(\log^{(q-1)/2}(qn^qm/\delta))$ with probability at least $1 - \delta/(20nm)$. If this occurs, then $\|y^\ell\|_2^2 = O(\log^{(q-1)}(qn^qm/\delta))\|x\|_2^2 = O(\log^{(q-1)}(qn^qm/\delta))$ since $\|x\|_2 = 1$. Taking a union bound over all $m$ vectors $v^\ell$ and all $n$ vectors $x^i$ we have that $\|A\|_2 = O(\frac{\log^{(q-1)}(qn^qm/\delta)}{m})$ with probability at least $1 - \delta/20$. $\quad\square$

**Lemma A.5.** *Let $x^i \in \mathbb{R}^{n^{q-1}}$ be an arbitrary vector and let $v^\ell = u^{\ell,2} \otimes u^{\ell,3} \otimes \cdots \otimes u^{\ell,q}$ be the tensor product of $q - 1$ random sign vectors $u^{\ell,j} \in \{-1, +1\}^n$. Then $|\langle v^\ell, x^i \rangle| = O(\log^{(q-1)/2}(qn^qm/\delta))$ with probability at least $1 - \delta/(20nm)$*

*Proof.* Without loss of generality, we can prove this for the case where $x^i$ is a unit vector. In this case it suffices to show that $|\langle v^\ell, x^i \rangle| = O(\log^{(q-1)/2}(qn^qm/\delta))\|x^i\|_2^2$ with probability at least $1 - \delta/(20nm)$. Proof by induction. The base case is $k = 2$: In this case $v^\ell = u^{\ell,2}$, so $v^\ell$ is a random sign vector. Applying Khintchine's inequality with $\delta' = \delta/(40n^2m)$ shows that $|\langle v^\ell, x^i \rangle| = O(\log^{1/2}(40n^2m/\delta))\|x^i\|_2^2$ with probability at least $1 - \delta/(40n^2m)$, so the base case holds. Assume that for $k = q - 1$, $|\langle v^\ell, x^i \rangle| = O(\log^{(q-2)/2}(qn^qm/\delta))\|x^i\|_2^2$ with probability at least $1 - \delta(q-1)n^{q-2}/(20qn^qm)$. In the case of $k = q$, note that by Lemma A.8, we have that

$$|\langle v^\ell, x^i \rangle| = |u^{\ell,2^T} X(u^{\ell,3} \otimes \cdots \otimes u^{\ell,q})|,$$

where we have rewritten the vector $x^i$ as an $n \times n^{q-2}$ matrix $X$. Let

$$u' = (u^{\ell,3} \otimes \cdots \otimes u^{\ell,q}).$$

Note that $Xu'$ is a vector of length $n$ where the $p$-th entry is equal to $\langle X_{i,*}, u' \rangle$, where $X_{i,*}$ is the $i$-th row of $X$. Computing $\langle X_{i,*}, u' \rangle$ is simply the $k = q - 1$ case, so by the induction hypothesis, we know that $\langle X_{i,*}, u' \rangle = O(\log^{(q-2)/2}(qn^qm/\delta))\|x^i\|_2^2$ with probability at least $1 - \delta(q-1)n^{q-2}/(20qn^qm)$. Taking a union bound we have that every entry of $Xu'$ is simultaneously bounded by $O(\log^{(q-2)/2}(qn^qm/\delta))\|x^i\|_2^2$ for $i = 1\ldots n$ with probability at least $1 - \delta(q-1)n^{q-1}/(20qn^qm)$. We now compute $|u^{\ell,2^T}(Xu')|$. Since $u^{\ell,2}$ is a random sign vector, we apply Khintchine's inequality with $\delta' = \delta/(20qn^qm)$ and have that $|\langle v^\ell, x^i \rangle| = |\langle u^{\ell,2}, (Xu') \rangle| = O(\log^{1/2}(qn^qm/\delta))\|(Xu')\|_2^2$ with probability at least $1 - \delta/(20qn^qm)$. $\|(Xu')\|_2^2 \le \sum_{i=1}^n O(\log^{(q-2)/2}(qn^qm/\delta))\|x^i\|_2^2$, so by another union bound $|\langle v^\ell, x^i \rangle| = O(\log^{(q-1)/2}(qn^qm/\delta))$ with probability at least $1 - \delta/(20nm)$. $\quad\square$

**Lemma A.6.** *Let $x^i \in \mathbb{R}^{n^{q-1}}$ be an arbitrary unit vector and let $v^\ell = u^{\ell,2} \otimes u^{\ell,3} \otimes \cdots \otimes u^{\ell,q}$ be the tensor product of $q-1$ random sign vectors $u^{\ell,j} \in \{-1, +1\}^n$. Then $\langle v^\ell, x^i \rangle^2 \leq t^{1/2} \|x^i\|_2^2$ with probability at least $1 - qn^{q-1} e^{-\Theta(t^{1/(2(q-1))})}$.*

*Proof.* Proof by induction. The base case is $k = 2$. In this case $v^\ell = u^{\ell,2}$, so $v^\ell$ is a random sign vector. Applying Khintchine's inequality with $\delta' = qn^{q-1} e^{-\Theta(t^{1/2(q-1)})}$ we have that $(\langle v^\ell, x^i \rangle)^2 \leq t^{1/2} \|x^i\|_2^2$ with probability at least $1 - 2n e^{-\Theta(t^{1/2})}$. Assume that for $k = q - 1$, $(\langle v^\ell, x^i \rangle)^2 \leq t^{1/2(q-2)} \|x^i\|_2^2$ with probability at least $1 - (q-1)n^{q-2} e^{-\Theta(t^{(q-2)/(2(q-1))})}$. In the case of $k = q$, note that by Lemma A.8, we have that

$$|\langle v^\ell, x^i \rangle| = |u^{\ell,2^T} X (u^{\ell,3} \otimes \cdots \otimes u^{\ell,q})|,$$

where we have rewritten the vector $x^i$ as an $n \times n^{q-2}$ matrix $X$. Let

$$u' = u^{\ell,3} \otimes \cdots \otimes u^{\ell,q}.$$

Note that $Xu'$ is a vector of length $n$ where the $p$-th entry is equal to $\langle X_{i,*}, u' \rangle$, where $X_{i,*}$ is the $i$-th row of $X$. Computing $\langle X_{i,*}, u' \rangle$ is simply the $k = q - 1$ case, so by the induction hypothesis, we know $(\langle v^\ell, x^i \rangle)^2 \leq t^{(q-1)/2(q-2)} \|x^i\|_2^2$ with probability at least $1 - (q-1)n^{q-2} e^{-\Theta(t^{(q-2)/(2(q-1))})}$. Taking a union bound, for each of the $n$ entries of $Xu'$ it simultaneously holds that $|\langle X_{i,*}, u' \rangle| \leq t^{(q-1)/2(q-2)} \|X_{i,*}\|_2^2$ with probability at least $1 - (q-1)n^{q-1} e^{-\Theta(t^{(q-2)/(2(q-1))})}$. We apply Khintchine's inequality again with $\delta'$ to bound $\langle u^{\ell,2}, (Xu') \rangle$. Thus with a second union bound we have that

$$
\begin{aligned}
\langle u^{\ell,2}, (Xu') \rangle &\leq t^{1/2(q-1)} \sum_{i=1}^n \|(Xu')_i\|_2^2 \\
&\leq t^{1/2(q-1)} \cdot t^{(q-2)/2(q-1)} \sum_{i=1}^n \|X_{i,*}\|_2^2 \\
&\leq t^{1/2} \|x^i\|_2^2
\end{aligned}
$$

with probability at least $1 - qn^{q-1} e^{-\Theta(t^{1/(2(q-1))})}$. $\qquad\square$

**Lemma A.7.** *With probability at least $1 - \delta/20$,*

$$\|A\|_F = O(1/\sqrt{m} + \log^{1/2}(1/\delta) \log^{(2q-3)/2}(m/\delta) \log\log(m/\delta)/m)$$

*Proof.* As in Lemma A.4, $A$ is block-diagonal, and the $\ell$-th block in $A$ can be written as $(1/m)y^\ell (y^\ell)^T$ where $y^\ell \in \mathbb{R}^n$ for each $\ell \in [m]$, and $y_i^\ell = \langle v^\ell, x^i \rangle$. We therefore have that

$$\|A\|_F^2 = \frac{1}{m^2} \sum_{\ell=1}^m \|y^\ell\|_2^4. \tag{8}$$

Note that for $\ell = 1, \ldots, m$, the $\|y^\ell\|_2^4$ are independent random variables. Further for each $\ell \in [m]$ and any $t > 0$, we have,

$$
\begin{aligned}
\Pr[\|y^\ell\|_2^4 \geq t] &= \Pr[(\sum_{i=1}^n \langle v^\ell, x^i \rangle^2)^2 \geq t] \\
&\leq \Pr[\exists i \in [n] \text{ such that } \langle v^\ell, x^i \rangle^2 \geq \sqrt{t} \|x^i\|_2^2]. \tag{9}
\end{aligned}
$$

To understand $\Pr[\langle v^\ell, x^i \rangle^2 \geq \sqrt{t} \|x^i\|_2^2]$, note that by A.6, we have that $(\langle v^\ell, x^i \rangle)^2 \leq t^{1/2} \|x^i\|_2^2$ with probability at least $1 - qn^{q-1} e^{-\Theta(t^{1/(2(q-1))})}$. Since $q$ is constant this is $1 - n^{q-1} e^{-\Theta(t^{1/(2(q-1))})}$. Plugging into (9), we have that

$$\Pr[\|y^\ell\|_2^4 \geq t] \leq n^q \cdot e^{-\Theta(t^{1/(2(q-1))})}. \tag{10}$$

465 Equipped with (10), we now analyze $S := \sum_{\ell=1}^{m} \|y^\ell\|_2^4$. For $j \geq 1$, let $S_j = \{\ell \mid 2^j \leq \|y^\ell\|_2^4 \leq$
466 $2^{j+1}\}$, and let $S_0 = \{\ell \mid \|y^\ell\|_2^4 \leq 1\}$. Then $S \leq 2 \cdot \sum_{j \geq 0} 2^j |S_j|$. Using (9), for each $j$,

$$
\begin{aligned}
\Pr[|S_j| > \frac{t}{2^j 100 j^2}] &\leq \binom{m}{t/(2^j 100 j^2)} \cdot e^{-\Theta(2^{j/(2(q-1))}) \cdot t/(2^j j^2)} \\
&\leq \left( \frac{m 2^j 100 j^2 e}{t} \right)^{t/(2^j 100 j^2)} \cdot e^{-\Theta(t/(2^{j(2q-3)/(2q-2)} j^2))} \quad (11)
\end{aligned}
$$

467 We will set $t \geq m$, and thus (11) becomes:

$$
\Pr[|S_j| > \frac{t}{2^j 100 j^2}] \leq 2^{ct/(2^j j) - c't/(2^{j(2q-3)/(2q-2)} j^2)}, \quad (12)
$$

468 where $c, c' > 0$ are absolute constants. For $j$ larger than an absolute constant $j_0$, (12) is just equal to

$$
2^{-\Theta(t/(2^{j(2q-3)/(2q-2)} j^2))}. \quad (13)
$$

469 This probability is maximized when $j$ is as large as possible. To control it, we define the event $\mathcal{G}$
470 that $\|y^\ell\|_2^4 \leq C \log^{2(q-1)}(m/\delta)$ for a sufficiently large constant $C > 0$. W.l.o.g., we also choose
471 $C$ so that $2^{j_1} = C \log^{2(q-1)}(m/\delta)$ for an integer $j_1$. Applying (10) and a union bound, and using
472 the fact that $d < 1/n^q$, we have that with probability $1 - \delta/40$, simultaneously for all $\ell \in [m]$,
473 $\|y^\ell\|_2^4 \leq C \log^{2(q-1)}(m/\delta)$, and so $\Pr[|\cup_{j > j_1} S_j| = 0] \geq 1 - \delta/40$. Also, $\sum_{j=0,\ldots,j_0} 2^j |S_j| \leq Cm$,
474 for a constant $C > 0$, with probability 1.

475 Consequently,

$$
\begin{aligned}
\Pr[S > t] &\leq \Pr[2 \cdot \sum_{j \geq 0} 2^j |S_j| > t] \\
&\leq \Pr[2 \cdot \sum_{j=j_0}^{\infty} 2^j |S_j| > t - Cm] \\
&= \Pr[\sum_{j=j_0}^{\infty} 2^j |S_j| > (t - Cm)/2] \\
&\leq \sum_{j=j_0}^{\infty} \Pr[|S_j| > \frac{(t - Cm)/2}{2^j 100 j^2}] \\
&= \sum_{j=j_1}^{\infty} \Pr[|S_j| > \frac{(t - Cm)/2}{2^j 100 j^2}] + \sum_{j=j_0}^{j_1} \Pr[|S_j| > \frac{(t - Cm)/2}{2^j 100 j^2}] \\
&\leq \Pr[|\cup_{j > j_1} S_j| > 0] + \sum_{j=j_0}^{j_1} \Pr[|S_j| > \frac{(t - Cm)/2}{2^j 100 j^2}] \\
&\leq \delta/40 + \sum_{j=j_0}^{j_1} \Pr[|S_j| > \frac{(t - Cm)/2}{2^j 100 j^2}] \\
&\leq \delta/40 + \sum_{j=j_0}^{j_1} 2^{-\Theta(t/(2^{j(2q-3)/(2q-2)} j^2))}
\end{aligned}
$$

476 where the first inequality uses that $S < 2 \cdot \sum_{j \geq 0} 2^j |S_j|$, the second inequality uses that
477 $\sum_{j=0,\ldots,j_0} 2^j |S_j| \leq Cm$ with probability 1, the third inequality uses the fact that if we did not
478 have $|S_j| > \frac{(t-Cm)/2}{2^j 100 j^2}$ then we would have $\sum_{j=j_0}^{\infty} 2^j |S_j| \leq (t - Cm)/2$, the fifth inequality uses
479 that $\Pr[|\cup_{j > j_1} S_j| = 0] \geq 1 - \delta/40$, and the final inequality uses (13), the definition of $j_0$, and that
480 we can assume $t - Cm = \Theta(t)$ if we choose $t > 2Cm$.

Finally, note that $\sum_{j=j_0}^{j_1} 2^{-\Theta(t/(2^{j(2q-3)/(2q-2)}j^2))}$ is equal to $2^{-\Theta(t/(2^{j_1(2q-3)/(2q-2)}j_1^2))}$. Recalling that $2^{j_1} = C\log^{2(q-1)}(m/\delta)$, this expression is equal to $2^{-\Theta(t/(\log^{2q-3}(m/\delta)\log^2\log(m/\delta)))}$. Setting

$$t = C'\log(1/\delta)\log^{2q-3}(m/\delta)\log^2\log(m/\delta) + 2Cm,$$

for absolute constants $C, C' > 0$ makes this expression at most $\delta/40$, which gives us our overall bound that $\Pr[\sum_{\ell=1}^m \|y^\ell\|_2^4 > C\log(1/\delta)\log^{2q-3}(m/\delta)\log^2\log(m/\delta) + 2Cm] < \delta/20$.

Plugging into (8), with probability at least $1 - \delta/20$,

$$
\begin{aligned}
\|A\|_F^2 &= \frac{1}{m^2}\sum_{\ell=1}^m \|y^\ell\|_2^4 \\
&\leq \frac{C\log(1/\delta)\log^{2q-3}(m/\delta)\log^2\log(m/\delta)}{m^2} + \frac{2c}{m}.
\end{aligned}
$$

Thus we have that, with probability at least $1 - \delta/20$,

$$\|A\|_F = O(1/\sqrt{m} + \log^{1/2}(1/\delta)\log^{(2q-3)/2}(m/\delta)\log\log(m/\delta)/m).$$

$\square$

**Lemma A.8.** *Let $a \in \mathbb{R}^n$, $b \in \mathbb{R}^k$, and $x \in \mathbb{R}^{nk}$ be arbitrary vectors. Define $X \in \mathbb{R}^{n \times k}$ to be $x$ written as a matrix, such that $X_{i,j} = x_{(i-1)k+j}$. Then*

$$\langle (a \otimes b), x \rangle = a^T X b.$$

*Proof.* We have that $\langle (a \otimes b), x \rangle = \sum_{i=1}^n a_i \langle b^T X_{i,*} \rangle$, where $X_{i,*}$ is the $i$-th row of $X$. Note that the $i$-th element of vector $Xb$ is $\langle b^T X_{i,*} \rangle$, and so $a^T X b = \sum_{i=1}^n a_i \langle b^T X_{i,*} \rangle$. Thus $\langle (a \otimes b), x \rangle = a^T X b$. $\square$

**Lemma A.9.** *Let $A \in \mathbb{R}^{n \times n}$ and $B \in \mathbb{R}^{n \times k}$ be arbitrary matrices and let $x \in \mathbb{R}^{nk}$ be an arbitrary vector. Then $\|(A \otimes B)x\|_2^2 = \|AXB^T\|_F^2$, where $X$ is $x$ written as a matrix, such that $X_{i,j} = x_{(i-1)k+j}$.*

*Proof.* It suffices to show a bijection between the entries of $(A \otimes B)x$ and the entries of $AXB^T$. Note that the $(i,j)$-th entry of the matrix $AXB^T$ is equal to $\langle (A_{i,*} \otimes B_{j,*}), x \rangle$, where $A_{i,*}$ is the $i$-th row of A and $B_{j,*}$ is the $j$-th row of B. The entry at position $((i-1)k+j)$ in the vector $(A \otimes B)x$ is also equal to $\langle (A_{i,*} \otimes B_{j,*}), x \rangle$, giving the bijection. $\square$

# B  Proof of Theorem 2.4

It suffices to show that, for any unit vector $x \in \mathbb{R}^{n^q}$,

$$\Pr[|\|Tx\|_2^2 - 1| > \epsilon] \leq \delta. \tag{14}$$

To show (14), we use the following shown in the proof of Lemma 40 of [14].

**Lemma B.1.** *(Proof of Lemma 40 of [14]) Let $T^i : \mathbb{R}^{n'} \to \mathbb{R}^{t'}$ be a CountSketch matrix, where $t' = O(\epsilon^{-2}/(q\delta))$. Then for any fixed matrix $X \in \mathbb{R}^{n' \times n}$,*

$$\Pr[\|T^i X\|_F^2 = (1 \pm \epsilon)\|X\|_F^2] \geq 1 - \delta/q.$$

*Proof.* Lemma 40 of [14] shows that $\mathbf{E}[\|T^i X\|_F^2] = \|X\|_F^2$ and $\mathbf{Var}[\|T^i X\|_F^2] \leq \frac{6}{m}\|X\|_F^4$. Applying Chebyshev's inequality, we have that

$$\Pr[|\|T^i X\|_F^2 - \|X\|_F^2| \geq \epsilon\|X\|_F^2] \leq \frac{6\|X\|_F^4}{t'\epsilon^2\|X\|_F^4},$$

and setting $t' = 6q\epsilon^{-2}/(\delta)$ proves the lemma.

$\square$

We show (14) by applying Lemma B.1 $q$ times, each time with $\epsilon$ replaced with $\epsilon/(4q)$. By Lemma A.9 we have that $\|Tx\|_2^2 = \|T^1 X(T^2 \otimes T^3 \otimes \cdots \otimes T^q)\|_F^2$, where $X \in \mathbb{R}^{n \times n^{q-1}}$ has its entries in one-to-one correspondence with the entries of $x$. By Lemma B.1, $\|T^1 X(T^2 \otimes T^3 \otimes \cdots \otimes T^q)\|_F^2 = (1 \pm \epsilon/(4q))\|X(T^2 \otimes T^3 \otimes \cdots \otimes T^q)\|_F^2$. Now we replace $X$ with the matrix $X^1 \in \mathbb{R}^{n \times (t \cdot n^{q-2})}$ which has each of its columns $X_{*,i}$ replaced with $T^1 X_{*,i}$. The entries of $X^1$ are then in one-to-one corresponding with the entries of a vector $x^1 \in \mathbb{R}^{t \times n^{q-1}}$.

We now repeat the above argument with $T^2$ replacing the role of $T^1$ and $X^1$ replacing the role of $X$. Applying Lemma B.1 $q$ times, and applying a union bound, we obtain a vector $Tx \in \mathbb{R}^{t^q}$ with $\|Tx\|_2^2 = 1 \pm \epsilon$ with probability at least $1 - \delta$. This proves (14).

Note that if the vector $x$ is of the form $x^1 \otimes x^2 \otimes \cdots \otimes x^q$, for $x^i \in \mathbb{R}^n$ for $i = 1, 2, \ldots, q$, then we can write $Tx$ as

$$
\begin{aligned}
Tx &= T^1 \otimes T^2 \otimes \cdots \otimes T^q (x^1 \otimes x^2 \otimes \cdots \otimes x^q) \\
&= T^1 x^1 \otimes T^2 x^2 \otimes \cdots \otimes T^q x^q
\end{aligned}
$$

Since each matrix $T^i$ is a CountSketch matrix, computing $T^i x^i$ takes $O(\sum_{i=1}^q \text{nnz}(x^i))$ time, and additionally $\text{nnz}(T^i x^i) \le \text{nnz}(x^i)$ for each $i = 1, 2, \ldots, q$.

## C  Proof of Theorem 3.1

Consider a vector $x \in \mathbb{R}^{n^q}$ defined as follows: $x = y \otimes y \otimes \cdots \otimes y$, where $y$ is a random sparse vector containing $(1/q)\log(1/(4\delta))$ entries that are equal to 1 placed at uniformly random positions, and remaining entries equal to 0. Note that $T^i$ perfectly hashes the $(1/q)\log(1/(4\delta))$ ones in $y$ with probability at least $1 - \delta \cdot \Theta(\log^2(1/\delta)/q^2)$. By a union bound, with probability at least $1/2$, simultaneously for $i = 1, 2, \ldots, q$, $T^i$ perfectly hashes $y$. Thus, conditioned on this event which we call $\mathcal{E}$, each $T^i y$ has exactly $(1/q)\log(1/(4\delta))$ entries which are each equal to 1 or $-1$, and remaining entries are equal to 0. We condition on event $\mathcal{E}$ in what follows.

Now consider the first row $u^{1,1} \otimes u^{1,2} \otimes \cdots \otimes u^{1,q}$ of the sketching matrix $\sqrt{m} \cdot S$. The first entry of $\sqrt{m} S \cdot Tx$ is equal to $\prod_{i=1}^q \langle u^{1,i}, T^i y \rangle$. For each $i = 1, 2, \ldots, q$, with probability $(1/2)^{(1/q)\log(1/(4\delta))} = (4\delta)^{1/q}$, each of the entries of $u^{1,i}$ in the support of $T^i y$ has the same sign. Thus, this holds for all $i$ simultaneously with probability $(4\delta)$ by independence of the $u^{1,i}$. Let us call this event $\mathcal{F}$. By independence of $S$ and $T$ it follows that $\mathcal{E} \wedge \mathcal{F}$ occurs with probability at least $(1/2) \cdot (4\delta) = 2\delta$. In this case, we have that the squared first entry of $\sqrt{m} \cdot STx$ has value $\Omega(\log^{2q}(1/\delta))$, where we again used that $q$ is a constant. Note that $\|x\|_2^2 = (1/q)^q \log^q(1/(4\delta))$, which for constant $q$, is a factor of $\Theta(\log^q(1/\delta))$ smaller than the squared first entry of $\sqrt{m} S \cdot Tx$ conditioned on $\mathcal{E} \wedge \mathcal{F}$.

We next consider $\|(STx)_{-1}\|_2^2$, which denotes the squared 2-norm of the vector $STx$ with the first entry replaced with 0. Define the event $\mathcal{G}$ that $\|(STx)_{-1}\|_2^2 = (1 \pm C/\sqrt{m})\|x\|_2^2$ for a constant $C > 0$ defined below, where $C$ may depend on $q$ but is constant for constant $q$, as we assume.

We will show that $\Pr[\mathcal{G} \mid \mathcal{E} \wedge \mathcal{F}] \ge 1/2$. We will then have that $\Pr[\mathcal{E} \wedge \mathcal{F} \wedge \mathcal{G}] \ge \frac{1}{2} \cdot 2\delta = \delta$. Since the first rows of $S$ are independent, $\Pr[\mathcal{G} \mid \mathcal{E} \wedge \mathcal{F}] = \Pr[\mathcal{G} \mid \mathcal{E}]$, which we now bound.

Note this will imply a lower bound of $m = \Omega(\epsilon^{-1} \log^q(1/\delta))$, since it implies that with probability at least $\delta$,
$$
\|STx\|_2^2 = (\Omega(\log^{2q}(1/\delta))/m + 1 \pm 10/\sqrt{m})\|x\|_2^2.
$$
Since $m \ge 10000/\epsilon^2$ by our $m = \Omega(\epsilon^{-2}\log(1/\delta))$ lower bound, and assuming $\delta$ is smaller than a sufficiently small constant, it follows that with probability at least $\delta$,

$$
\|STx\|_2^2 = (\Omega(\log^q(1/\delta))/m + 1 \pm \epsilon/10)\|x\|_2^2. \tag{15}
$$

In order for $\|STx\|_2^2 = (1 \pm \epsilon)\|x\|_2^2$ with probability at least $1 - \delta$, we must therefore have $m = \Omega(\epsilon^{-1}\log^q(1/\delta))$, which shows the lower bound.

Thus, it remains to show that $\Pr[\mathcal{G} \mid \mathcal{E}] \ge 1/2$. Note that $\|Tx\|_2 = \|x\|_2$ given that event $\mathcal{E}$ occurs, and more precisely $\|T^i y\|_2 = \|y\|_2$ for each $i = 1, \ldots, q$, so it suffices to compute the probability that $S$ preserves the norm of $Tx = T^1 y \otimes T^2 y \otimes \cdots \otimes T^q y$. For the $\ell$-th row $u^{\ell,1} \otimes u^{\ell,2} \otimes \cdots \otimes u^{\ell,q}$ of $S$, we have $\sqrt{m}(STx)_\ell = \prod_{i=1}^q \langle u^{\ell,i}, T^i y \rangle$. Define $z_\ell = \prod_{i=1}^q \langle u^{\ell,i}, T^i y \rangle$.

Since the $u^{\ell,i}$ are independent for $i = 1, 2, \ldots, q$, we have

$$
\begin{aligned}
\mathbf{E}[z_\ell^2] &= \prod_{i=1}^{q} \mathbf{E}[\langle u^{\ell,i}, T^i y \rangle^2] \\
&= \prod_{i=1}^{q} \|T^i y\|_2^2 = \prod_{i=1}^{q} \|y\|_2^2 = \|x\|_2^2.
\end{aligned}
\tag{16}
$$

Consequently, $\mathbf{E}[\|(STx)_{-1}\|_2^2] = \frac{m-1}{m}\|x\|_2^2$.

We can similarly bound the second moment,

$$
\mathbf{E}[z_\ell^4] = \prod_{i=1}^{q} \mathbf{E}[\langle u^{\ell,i}, T^i y \rangle^4],
\tag{17}
$$

where we have again used independence of the $u^{\ell,i}$ for $i = 1, 2, \ldots, q$. Note that $\mathbf{E}[\langle u^{\ell,i}, T^i y \rangle^4]$ is just the second moment of the standard Alon-Matias-Szegedy [5] estimator (using a random sign vector $u^{\ell,i}$) for the squared 2-norm of a fixed vector (in this case $T^i y$), and it holds (see the proof of Theorem 2.2 of [5]),

$$
\mathbf{E}[\langle u^{\ell,i}, T^i y \rangle^4] \le \|T^i y\|_4^4 + 6\|T^i y\|_2^4 \le 7\|T^i y\|_2^4.
\tag{18}
$$

Plugging (18) into (17), we get

$$
\begin{aligned}
\mathbf{E}[z_\ell^4] &\le 7^q \prod_{i=1}^{q} \|T^i y\|_2^4 \\
&= 7^q \left(\prod_{i=1}^{q} \|T^i y\|_2^2\right)^2 = 7^q \|x\|_2^4.
\end{aligned}
\tag{19}
$$

Consequently by (19), $\mathbf{Var}[z_\ell^4] \le 7^q \|x\|_2^4$, and by independence of $\ell = 2, 3, \ldots, m$, $\mathbf{Var}[\|(STx)_{-1}\|_2^2] \le \frac{1}{m-1} \cdot 7^q \|x\|_2^4$. Combining with (16), we can apply Chebyshev's inequality to conclude that

$$
\begin{aligned}
&\Pr[|\|(STx)_{-1}\|^2 - \|x\|_2^2| > \gamma \|x\|_2^2] \\
&\le \frac{7^q \|x\|_2^4}{(m-1)\gamma^2 \|x\|_2^4} = \frac{7^q}{(m-1)\gamma^2}.
\end{aligned}
\tag{20}
$$

It follows from (20) that for constant $q$ and $\gamma = \Theta(1/\sqrt{m})$, this probability is at least $1/2$. Here we can take the constant $C$ defined above to be $2 \cdot 7^{q/2}$, for example. Thus, $\Pr[\mathcal{G} \mid \mathcal{E}] \ge 1/2$, which completes the proof.

## D   Note on Previous Analysis of Sketch

Given $Sx$ and $Sy$ for $x, y \in \mathbb{R}^{n^q}$, it is not hard to show that $\mathbf{E}[\langle Sx, Sy \rangle] = \langle x, y \rangle$. The main issue is the variance of this sketch. Indeed, as stated in [34], this estimate "incurs very large variance, especially for large $q$". Kar and Karnick analyze the variance of this sketch and show the following (discussion before Section 4.1 of [34]). Suppose one has a set $\Omega$ of points of the form $a^{\otimes q}$ for some $a \in \mathbb{R}^n$ (the different points in $\Omega$ may be tensor products of different points $a \in \mathbb{R}^n$), for which each such point $a$ is in the radius-$R$ $\ell_1$-ball $B_1(0, R)$. Let $C_\Omega = q(qR^2)^q$. Then if $m = \Omega(C_\Omega^2 \epsilon^{-2} \log(1/\delta))$, then for any $x, y \in \Omega$,

$$
\Pr[|\langle Sx, Sy \rangle - \langle x, y \rangle| > \epsilon] \le \delta.
$$

For $a = (1/\sqrt{n}, \ldots, 1/\sqrt{n})$, we have $\|a^{\otimes q}\|_2 = 1$ but $\|a^{\otimes q}\|_1 = n^{q/2}$. Consequently, to apply their analysis we would need to set $R = n^{1/2}$ in their bound, which gives $C_\Omega = n^{2q}$ and a sketching dimension $m = \Omega(n^{2q}\epsilon^{-2}\log(1/\delta))$ which is much larger than the dimension $n^q$ of $a^{\otimes q}$ to begin with!