[Reviews · NeurIPS 2019]

Reviewer 1



This work achieves an improved bound on the sample complexity of random tensor projection and it is argued that this bound is tight and nearly optimal. A key observation is to view the random sketch as a bilinear form of a random matrix. It makes the analysis suitable to apply general matrix concentration inequalities. The authors can obtain better bounds by analyzing both operator and Frobenius norm of the random matrix, which is the key challenges of this work. Their proof techniques are different from previous approaches but very impressive. It has a great impact in term of applying complex matrix concentration inequalities for exponentially improved bound. This work can give a good direction to challengeable analysis in many other works. The paper is also well-written and its organization is comprehensive. Although the proof techniques can be complicated and require a large amount of prior knowledge, the contributions and proofs are straightforward to understand. However, readers not familiar with prior works of random tensor sketch may have some hardness since there is no kind preliminaries section. It would be better to provide some backgrounds with a concrete description including Johnson-Lindenstrauss transform, CountSketch, and TensorSketch. In overall, I believe that the impact of this work is impressive and vote for acceptance. Some minor issues or typos: - Please write the detail information of reference [28]. If the reference is not published yet, it would be better not to refer it. - ln equation between line 135 and 136, the notation of norm is not complete. - In line 191, the right parenthesis is missing. - In below the equation (2) (between line 155 and 156), S^i x^i should be vector while the right-hand side seems to be scalar. ============================================================================== I read all reviews and author feedback. The final manuscript with authors response can improve the writing quality and highlight the novelty of this work.

Reviewer 2



Summary: The paper presents a sketching method (a variant of the method given in Kar and Karnick, 2012) for tensors and provides its theoretical analysis. Specifically, the authors improve the previously known rate for the sketch-size n^q (exponential in data dimension n) to log^q(n). Further, the authors combine CountSketch with the proposed method, which enjoys a better theoretical guarantee (the dependence on n is removed) while fast for rank-1 tensors (the computational cost given by the order of sketch size * nnz(x^i), x^i is a n-dim component vector of a q-th order tensor). It is shown that the proposed method outperforms TensorSketch (Pham and Pagh, 2013) and RandomMaclaurin (Kar and Karnick, 2012) in approximating polynomial kernels. Also, the proposed method has a better test-error performance in compressing a neural network compared to Arora et al., 2018 with a small sketch size. The analysis given by the authors is technically sound. Tensor sketching is an important task in machine learning, and the exponential improvement of the sketch size is useful particularly in high-dimensional settings such as NLP (e.g. a vocabulary set for text representation could be large). I also value the practicality of the proposed method using CountSketch as sketching itself could be computationally expensive. Together with the results in the experiments, I consider the contributions given by the paper significant. Some clarifications that could make the paper more accessible: * The scaling given in Kar and Karnick, 2012 is different from the proposed method due to the coefficient by Maclaurin. Should not this be clarified? " Line For each sketch coordinate it randomly picks degree t with probability 2−t and computes degree-t Tensorized Random Projection." What does degree mean here? Minor points: * The statements regarding epsilon-distortion in Theorem 2.1, 2.2, and Lemma 2.3: should they not use equality? e.g. P(1-eps||x|| <= ||Sx|| <= 1+eps||x||) * The scaling 1/\sqrt{m} is missing in line 151 (S^i) and 168 (S_0). * Should the proof lemma clarity that it deals with the case where x^i is a unit vector? * Line 437: p-th entry -> i-th entry? * Lemma 4.6 Inductiion index should be q instead of k (i.e. prove for q=2, and then q=k>2?)

Reviewer 3



This paper has studied theoretical property of existing Tensorized Random Projection. It has also provided a study of theoretical property of sketching method by combining tensorized Random Projection with a CountSketch. Theoretical bound improvement is marginal. More over no experimental comparison has been done to compare the performance of combine method and existing Tensorized Random Projection. Paper is technically sound and has theoretical significance. But is not well written and it needs improvement. Major comments: Authors have provided proof of having an exponentially better embedding dimension for a particular selection of S in Theorem 2.2. It is not clear if that particular choice of S is a contribution to this paper. If so, then it is better to provide a clear introduction of the proposed S before theorem 2.2. What is T is in Theorem 2.4 is not clear to me? It is not clear if experiments have been done with existing Tensorized Random Projection which has been proposed by [29] or some new version of it which is different than the existing [29] one. If it is the Tensorized Random Projection from [29] then motivation behind the experiments is not clear to me. If it is something different than Tensorized Random Projection from [29] then it is better to have a comparison with Tensorized Random Projection from [29]. Figure1 is not legible. What happens if m=n for both the methods? Minor comments: The required definition of few notations has not been provided. For example, in line 48 the definition of sketching matrix is not clear. Why all experiments have been provided for a polynomial kernel of degree 2? What happens if we use higher-order kernels? ---After Rebuttal--- the author has replied my comments. Inclusion required definitions have been promised. I would like to increase my score to 6 provided they update the manuscript as promised in rebuttal.

[Author Response · NeurIPS 2019]

We thank all three reviewers for their insightful and constructive comments. Please find our detailed response below.

**Reviewer 1** We appreciate the concrete suggestions for refining the presentation further. We are definitely going to add a
subsection that describes technical background and key tools such as Johnson-Lindenstrauss transforms or CountSketch
matrices to make the text even friendlier for the general reader.

The main benefit of Tensorized Random Projection (TRP) compared to TensorSketch (TS) is that TRP achieves high
probability bounds unlike TS, which provably does not. Section 4.1 discusses and demonstrates this. We'll emphasize it
more explicitly in the introduction to make it more prominent. Minor issues and typos:
• We'll delete references to unpublished [4, 28].
• Thanks for catching next two typos, fixed in our copy.
• *"(between line 155 and 156), $S^i x^i$ should be vector while the right-hand side seems to be scalar"*: Indeed, it should
be: Note that $(S^i x^i)_\ell$, the $\ell$th coordinate of $S^i x^i$, is $(1/\sqrt{m}) u_i^{\ell,1} \langle v^\ell, x^i \rangle$. The rest of the proof continues as before.

**Reviewer 2** Thanks again for reading the proofs carefully and for the helpful suggestions.

Clarifying *"The scaling given in Kar and Karnick ... For each sketch coordinate it randomly picks degree $t$ ..."*: We
believe both questions refer to lines 265-267, where we only meant to summarize Algorithm 1 on page 6 of Kar
and Karnick really briefly. Scaling by Maclaurin coefficients was omitted by mistake, and will be addressed in the
next version. Integer $t$ in the second sentence is denoted by $N$ in the original Kar and Karnick paper [29]. The first
occurrence of the word degree referred to the exponent of a term in the Maclaurin series and the second referred to the
order of the tensor created by raising the dot product of two vectors to the $t$th power. We'll rewrite and expand the
description of Kar and Karnick's method to avoid any ambiguity. Minor points:
• $a = (1 \pm \epsilon)b$ form is common in our experience, nevertheless we'll replace these statements with the canonical
$(1 - \epsilon)b \le a \le (1 + \epsilon)b$ form.
• Thanks for spotting the 4 typos, all are fixed.

The current proof of Theorem 2.1 requires that the sketch T maps to $t = \Theta(q^3/(\epsilon^2 \delta))$ intermediate sketching dimensions.
$S \cdot T$ is an improvement over S if $t$ is less than input dimension $n$. $\delta$ was small in our experiments and $n \le 100$ usually,
except for MNIST where $n = 784$. Thus $t$ would be much higher than the $n$ dimensions we started with. The running
time of $S$ is already quite fast on sparse data. Nevertheless we could try $S \cdot T$ with a heuristic dimension set for T in the
next version; thanks for suggesting it.

**Reviewer 3** Thank you for the detailed review. Major comments:
• $S$ is currently defined in lines 156-149, we'll pull that into a standalone definition preceding Theorem 2.2.
• Precise mathematical definition of $T$ given in Theorem 2.4. It can be rephrased as follows: $T$ sketches input tensor
$x^1 \otimes x^2 \otimes \cdots \otimes x^q$ by applying a CountSketch, see [13], $T^i$ to each $x^i$ independently and outputs tensor product of
$T^i x^i$. CountSketch of vector $v \in \mathbb{R}^n$ is $w \in \mathbb{R}^t$ such that $w_j = \sum_{i \in [1,n]:h(i)=j} v_i \cdot r_i$, where $h : [1,n] \to [1,t]$ is a
hash function and $r_i$ are iid $\pm 1$ random variables.
• Kar and Karnick [29] proposed the Random Maclaurin (RM) sketch, and implicitly, without naming it as such,
introduced Tensorized Random Projection (TRP) as a building block of RM. The two sketches are different. Figure 2
compares TRP and RM and demonstrates that TRP is vastly more accurate. RM expands the kernel into its Maclaurin
series, randomly chooses a Maclaurin term, and uses TRP as a building block to estimate that term. A simple yet
important message of our paper is that for polynomial kernels one does not need the Maclaurin series expansion
because for the homogeneous kernel $\langle x, y \rangle^q$ exactly one Maclaurin coefficient equals 1 and all the others are 0. When
the latter are chosen by RM the sketch is the constant 0, wasting space. The inhomogeneous kernel $(1 + \langle x, y \rangle)^q$ is
best handled by augmenting the input with a constant 1 feature reducing it to the homogeneous case. Prior work
[29, 34] somehow overlooked these facts and ran experiments only with RM. Our main theoretical contribution is an
exponential improvement in the sketching dimension of a previously considered TRP. Prior work [29, 34] claimed
that TRP required many more dimensions (discussed in Section D) and thus was much less practical. Our work shows
that same exact sketch in fact has a much better complexity than was known. This also enables new applications of
the TRP sketch, e.g., to neural networks, that might not have previously been considered given that people thought
the behavior of this sketch was worse than it actually is.
• We'll increase font size, weight and line width of Figure 1 to improve contrast and repeat in its title that error bars
correspond to one standard deviation. With regard to the $m = n$ case, for $m = 100$ TensorSketch's error is always
the maximum possible, 1 at $n = m$, whereas the error of TRP is much lower and increases very slowly from about
$0.37$ at $n = 40$, where we truncated Figure 1(a), to about $0.39$ at $n = m = 100$. It's still at most $0.41$ at $n = 200$.
We'll explain this in the text and plot a broader range.
Minor comments:
• Line 48: We'll define dot-product preserving sketching matrices before referring to them.
• Polynomial kernel of degree 2 is one of the most popular, degrees 3 and 4 are also used in practice, higher degrees
are rare. We ran experiments with degrees 3 and 4 as well, the results were qualitatively similar. We'll include these
in the final version.

[Meta-Review · NeurIPS 2019]

This paper presents tight bounds on the dimension of random projection for tensor product of vectors, achieving exponential improvement on sketch dimension compared to the prior work. Their method also enjoyes efficient computation. The reviewers found the work solid and of high significance.